# Effects of Site Karst Water on Fresh Coal Gangue at Baizhuang Coalfield, China—Leaching Characteristic

**Bin-Bin Jiang [1], Yu-Kai Huang [2,\*], Dong-Jing Xu [3,\*], Zhi-Guo Cao [1] and Min Wu [1]**

1   State Key Laboratory of Water Resource Protection and Utilization in Coal Mining, Beijing 100083, China
2   National Energy Group Baorixile Energy Co., Ltd., Chenqi, Hulunbuir 021000, China
3   College of Earth Sciences and Engineering, Shandong University of Science and Technology,
    Qingdao 266590, China
*   Correspondence: nmhyk@126.com (Y.-K.H.); xudongjinggg@126.com (D.-J.X.)

**Abstract:** In mining areas where gangue is used for infill mining, Ordovician limestone karst water is affected by discarded gangue, and the water quality changes significantly. In this study, the effects of gangue on water quality change under different immersion solution conditions were evaluated by Zhai Zhen Coal Mine, Hongqi Coal Mine Ordovician Limestone Karst Water, and Baizhuang Coal Mine. The results showed that the cations in each group of immersion solutions had similar trends, and the $Na^+$ concentration fluctuations were greater in karst water immersion solutions with higher initial sodium ion concentrations, while the fluctuations of calcium and magnesium ions were more obvious in the early stage of immersion, and the fluctuations were gradually slowed down in the later stage. The anions in the immersion solution also have a similar fluctuation trend, but only in the early stage of immersion. By comparing the changes of the three indexes (pH, TDS, and ORP) in the immersion solution, it was found that highly mineralized karst water had a good buffering effect on the changes in the basic water quality index, while the ultrapure water quality index with low salinity had the greatest change. The 100% stacked columnar plot between the main water chemical ion changes shows the relative scale relationship of the water chemical components in the immersion solution at each period, and the water chemical components change significantly in the early immersion period (0–7 d), and then enter the fluctuation changes period, and the ion changes in the fluctuation changes period are mainly $Na^+$ and $SO_4^{2-}$ ions. This study provides a theoretical basis for exploring the changes of gangue to the quality of karst water in Ordovician limestone and also provides theoretical guidance for the study of groundwater pollution mechanisms in closed coal mines.

**Keywords:** immersion test; water quality; potential pollution components; site karst water





## 1. Introduction

Under the background of China's existing "rich coal, poor oil, less gas" energy structure and "double carbon", coal resources will continue to be China's main energy source and important industrial raw materials for a long time to come [1,2]. How to improve the energy efficiency of coal resources and reduce their impact on the ecological environment and achieve efficient development of greening and low carbon has become a major focus today. In order to ensure the scientific and rational exploitation of coal resources and transform the development mode of the existing coal industry, China has successively introduced relevant policies to vigorously integrate coal resources. In order to accelerate the transformation of the development mode of the coal industry, China will continue to restrict the eastern region, control the central and northeastern regions, and optimize the development of coal resources in the western region. During the "13th Five-Year Plan" period, China has withdrawn about 5500 coal mines and withdrawn backward coal production capacity of more than 1 billion tons per year. By the end of 2020, the number of coal mines

in China will be reduced to less than 4700, and the average annual production capacity of a single well (ore) will be increased to more than 1.1 million tons [3]. It is estimated that by the end of the 14th Five-Year Plan, domestic coal production will be controlled at about 4.1 billion tons, and China's coal consumption will be controlled at about 4.2 billion tons. The number of coal mines in the country is controlled at about 4000, 3–5 world-class coal enterprises with global competitiveness are cultivated, and the utilization of coal gangue and mine water and the discharge rate of the standard are 100% [4].

Long-term coal mining has caused many hydrogeological and ecological environmental problems, and at the same time, a large amount of mine water has been produced, which not only causes damage to the environment after direct discharge of mine water, but also seriously restricts the production of mines and affects the lives of local residents [5–7]. The types of mine water in China are roughly divided into six types: highly mineralized mine water, highly sulfate mine water, high fluoride mine water, acid mine water, common component mine water, and mine water with special components [8]. Theoretical research and practice show that gangue is mainly composed of carbon, hydrogen, oxygen, sulfur, iron, aluminum, silicon, calcium, and other macro elements and a variety of trace elements [9]. The potentially harmful elements, such as Fe, Zn, Mn, Cu, Pb, Cd, As, etc., are usually the polluting components that have the most serious impact on human domestic water in coal-based deposits, especially closed pit coal mines [10,11]. However, its main source is the solid waste remaining in the goaf, as the most important solid waste gangue after coal mining, it is left in the goaf area and all corners of the well lane, and with the wide application of filling mining and a large amount of gangue filled in the goaf, gangue and mine water adsorption, dissolution, and precipitation; ion exchange; and oxidation, reduction, and other effects, to a certain extent, affect and change the original water quality of mine water [12,13].

Shandong has typical characteristics of Carboniferous-Permian coal-bearing strata in North China and has special karst groundwater conditions, which makes the groundwater pollution mechanism of abandoned coal mines in our province very different from the previous research model [14]. Moreover, the intensity and depth of coal mining in our province are extremely large, and nearly half of the mines with a depth of more than 1000 m in the country are distributed in Xinwen, Feicheng, Yanzhou, Juye and other mining areas. Therefore, the combination of complex geological environments such as high water temperature and high karst water pressure, as well as many factors, such as environmental sensitivity and greater fluidity of karst groundwater, further aggravates the risk of karst water pollution in closed pit mines. What is more worrying is that, at present, the water supply in most of the closed pit mines in Shandong Province is in short supply; however, at present, the environmental impact mechanism of coal gangue on karst groundwater in the closed pit coal mine and the pollution indicators that need to be paid attention to are rarely studied, which has become a key problem that needs to be solved urgently in many mining areas.

This research selected typical closed coal mines in Shandong Province as the study area and investigated the effects of different types of site karst water of using coal gangue as the experimental material with regard to characteristics of conventional hydrochemical ions and to compare its differences from other water quality of immersion solutions. The changes of the main water chemical indexes (pH, TDS, and ORP) in each group immersion solutions were studied, and the main factors affecting the change of water quality indexes were explored. The results are intended to serve as a reference for the pollution evaluation of Ordovician limestone waters present in closed coal mines and forecasting of the ecological risks induced by coal gangue and to examine the safety of applying coal gangue to the goaf. These also provide theoretical guidance for the study of groundwater pollution mechanism in closed coal mines.

## 2. Materials and Methods

### 2.1. Sampling and Preparation

The collection of samples involve three coal mines (seen in Figure 1), namely the Baizhuang coal mine, the Zhaizhen coal mine, and the Hongqi coal mine. The coal-bearing strata of these coal mines are all Carboniferous-Permian in North China, and coal resources are exploited under aquifer in Ordovician limestone. Fresh coal gangue samples from the Hongqi coal mine were collected as the main experimental solid samples.

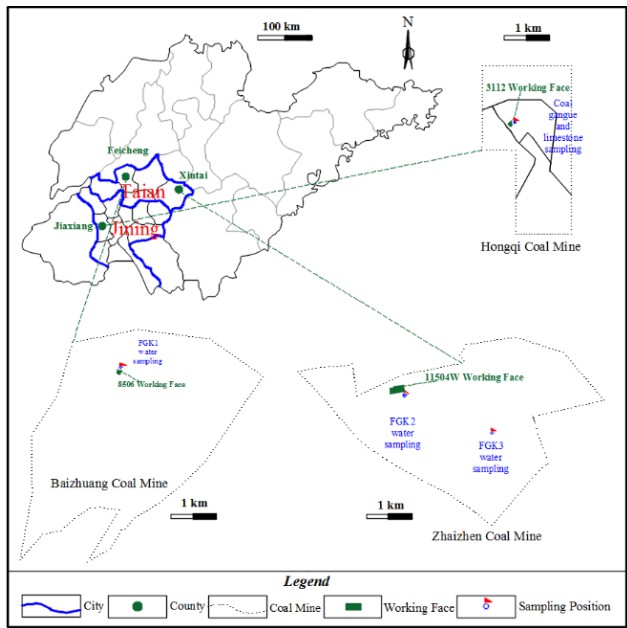

**Figure 1.** Location of the area.

The snake point distribution method and polypropylene shovel were used to collect coal gangue at 15 points in the 3112 working face at a burial depth of 358 m. This was done after removing the surface dirt from the sample and immediately storing it in a clean and sealed polypropylene bag for decolorization. Next, a process of sample mixing and crushing by crusher was conducted, and then the coal gangue with diameter size of 0.45~3.2 mm was sieved to homogenize. At last, coal gangue samples (roughly 1.5 kg) were weighed, wrapped with tin foil paper, and placed in a drying dish for subsequent immersion test.

The specific details of water samples in immerse test are as follows:

Water sample 1: Ordovician limestone karst water with water level of −250 m and water pressure of 1.9 MPa from the hydrological hole of 8506 working face in Baizhuang coal mine was selected.

Water sample 2: Ordovician limestone karst water with water level of −549.8 m and water pressure of 2.8 MPa from the hydrological hole of 1504 w transportation roadway in Zhaizhen coal mine was selected.

### 2.2. Static Immersion Test

Static immersion test is a commonly used method to evaluate the release potential of chemical components of solid materials. It can not only effectively solve the problem that it is difficult to collect water samples in closed mine goafs, but also facilitate the simulation of field conditions [15–18]. Moreover, the initial concentration of different chemical components in water and the release law over time are analyzed by changing the experiment parameters. So far, many scholars have done a lot of work on water pollution caused by wastewater discharged during the process of coal mining [19], but there is less

research about the impact of coal gangue on the quality of Ordovician limestone karst water caused by closed coal mines [11].

According to some scholars' research results about the proportional relationship between the coal gangue and the restored mine water [20], this immersion test between fresh coal gangue and Ordovician limestone karst water was conducted at a solid/liquid ratio of 1:10 (g/mL). Equal amounts of coal gangue samples were weighed and put into conical flask having cork with three immerse solutions, including one site karst water in Baizhuang, one site karst water in Zhaizhen, and ultrapure water (marked GK1, GK2, and GU, respectively, as shown in Table 1). Initial water quality parameters and hydrochemical composition of different immersion solutions can be seen in Table 1, and the ultrapure water was used as a contrast test. In order to realistically simulate the on-site temperature conditions of the goaf, the conical flask mixed with coal gangue and water was left to stand in a 40 °C constant temperature incubator. After soaking for 1 d, 2 d, 4 d, 7 d, 15 d, 25 d, and 35 d, 3 samples of each kind of immerse solutions were taken out and used as a parallel test, and a total of 9 test solutions for three types was sampled and analyzed. The leaching solutions were filtered into polyethylene bottles using a 0.45 μm aperture microporous membrane. Subsequently, the pH, ORP, EC, TDS, and DO after immersion were measured by a Hydrolab multi-parameter water quality analyzer. Using the Agilent's 7500 inductively coupled plasma mass spectrometer and the ICS-600 ion chromatograph of the Thermo-Fisher company in the American to measure the content of main cations (potassium, sodium, calcium, magnesium, etc.) and the anions (chloride, sulfate, nitrate, etc.), respectively.

**Table 1.** Initial Water Quality Parameters and Hydrochemical Composition of the Coal Gangue Immersion Tests under Different Immersion Solutions.

| Serial Number | Initial pH | Initial TDS (mg/L) | Initial ORP (mV) | $Na^+$ | $K^+$ | $Ca^{2+}$ | $Mg^2$ | $Fe^{2+}/Fe^{3+}$ | $Cl^-$ | $SO_4{}^{2-}$ | $NO_3{}^-$ | $F^-$ |
|---|---|---|---|---|---|---|---|---|---|---|---|---|
| | | | | | | | | (mg/L) | | | | |
| GK1 | 7.64 | 480.3 | 176.6 | 50.8 | 1.3 | 137.8 | 52.3 | 0.3 | 68.3 | 114.4 | 12.2 | 0.3 |
| GK2 | 7.69 | 1251.7 | 181.1 | 1000.0 | 16.4 | 217.5 | 69.1 | 0.7 | 82.6 | 1224.6 | 3.5 | 3.0 |
| GU | 5.83 | 2.9 | 277.2 | 0.0 | 0.0 | 0.0 | 0.0 | 0.0 | 0.0 | 0.0 | 0.0 | 0.0 |

## 3. Results and Discussion

### 3.1. Chemical Compositions of Coal Gangue

According to XRD analyses seen in Figure 2, the main minerals of coal gangue are kaolinite and quartz, and its secondary minerals are kaolinite, ankerite, smectite, sanidine, illite, and muscovite. This result is consistent with the analysis report of Ju-ye coal exploration. In addition, according to the geological data of Juye Coal field [21], this mine contains pyrite, but the possible content is less than 3% and has not been detected by XRD.

Through X-ray Fluorescence Spectrometry (XRF) at the Shandong University of Science and Technology, we can see that the contents of chemical composition in the coal gangue sample were $SiO_2$ (33.36%), $Al_2O_3$ (22.87%), CaO (22.56%), MgO (9.58%), $Fe_2O_3$ (5.63%), $K_2O$ (2.72%), $TiO_2$ (1.96%), $SO_3$ (0.66%), $Na_2O$ (0.45%), and MnO (0.08%). As seen from Figure 2, the column chart shows the major constituents from the original coal gangue sample were successively observed as $SiO_2$ > $Al_2O_3$ > CaO > MgO > $Fe_2O_3$. The ratio of $SiO_2/Al_2O_3$ in the coal gangue samples is larger than the theoretical ratio in kaolinite (1.18) [22], which indicated that the tested coal gangue samples mainly constitute kaolinite minerals and quartz, which is consistent with the XRD results.

The mine water studied in this paper is affected by the soaking of coal gangue, among which several water chemical components with large variations are sodium ion, calcium ion, magnesium ion, potassium ion, and iron ion. In addition, several types of oxides, such as CaO, MgO, $Fe_2O_3$, $K_2O$, and $Na_2O$, which are easy to react with water, including dissolved adsorption and ion exchange with water; they have a greater impact on water

quality—CaO is 8 times and 50 times that of K$_2$O and Na$_2$O, respectively, and MgO is 3 times and 20 times that of K$_2$O and Na$_2$O, respectively, with a large difference.

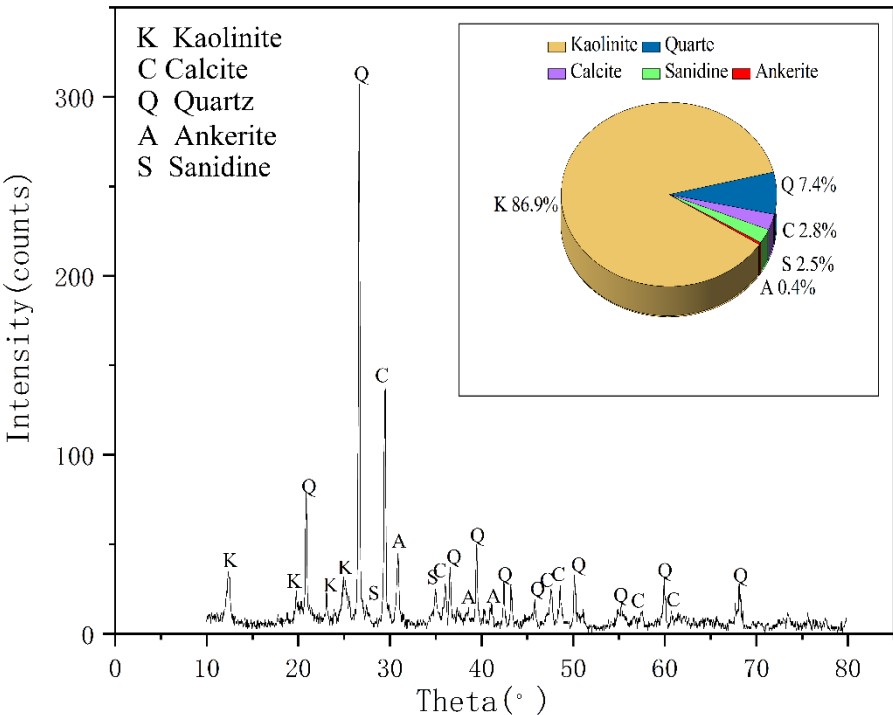

**Figure 2.** The mineral XRD analysis results plot.

*3.2. Main Hydrochemical Composition Variation under the Effects of Different Karst Water Solutions*

A massive immersion experiments of coal gangue were carried out under the conditions of site karst water and lab-configured water (GK1, GK2, and GU). Through ICP atomic emission spectrometry (ICP-AES), we obtained the concentrations of main hydrochemical composition including those of Na, K, Ca, and Mg in different solutions over time depicted in Figure 3. The concentration curves of all tested and highly affected major cations in the different solutions showed roughly similar trends after experiencing the largest fluctuation in the earlier stage. Each curve could be divided into three stages, including the initial violent fluctuating stage (0–7 d), the middle stable stage (7–25 d), and the last slow reaction stage (25–35 d). Especially, there were completely opposite change tendency between the dissolved Na cation concentrations and Ca, Mg, and K cations in different solutions within 4–7 d of the first stage, which indicated a transitional period of cations exchange between the dissolved Na and Ca, Mg, and K concentrations. Furthermore, the concentrations of main cations including those of Na, K, Ca, and Mg in solution GK2 with original high salinity kept the highest solubility over time, as depicted in Figure 3.

The dissolved Na levels in the different solutions over time were always ordered as GK2 > GK1 > GU, as seen in Figure 4a. The change curves of the Na concentrations in GK2 solutions were very similar, and the concentrations decreased dramatically to the relatively lower level (389.4 mg/L) within 0–4 d, and then increased quickly to the first little dissolved peak values (511.2 mg/L) after 7 d, then decreased moderately to the almost lowest values (242.9 mg/L) after 15 d, and then with a slight downward to the lowest value (223.1 mg/L) between 15 and 25 d, and then finally increased moderately to a relatively higher dissolved values (612.8 mg/L) next to the highest value in the final stage between 25 and 35 d. The Na concentrations for solution GK1 showed a similar developing trend as that of GK2, except rapid increase from 50.8 mg/L to 141.8 mg/L between 0 and 1 d. However, the Na concentrations for solutions GU and GK1 also showed a similar developing trend with each other. Within 0–1 d, the Na concentrations in solutions GU first quickly rose

from 0 to 48.4 mg/L, then continued to increase slowly to the first dissolved peak value (56.9 mg/L) after 7 d, next remained relatively stable from 7 to 25 d, and finally increased slowly from 30.2 mg/L to 97.7 mg/L within 25–35 d. From these results, we confirmed that the change trend in Na concentrations for site karst water fluctuated more greatly than that for lab-configured water, indicating that the effects of site karst water on Na in coal gangue obviously differed from lab-configured water. In combination with the detectable TDS content in the immersion solutions (ordered by GK2 > GK1 > GU) and the Na concentrations having the highest value of all cations, we inferred that sodium ions as the major element in the water solutions may control the change of the TDS content in the immersion solutions, although the dissolution, exchange, and precipitation of the ions in original water solutions and the minerals in coal gangue tried to maintain a dynamic balance.

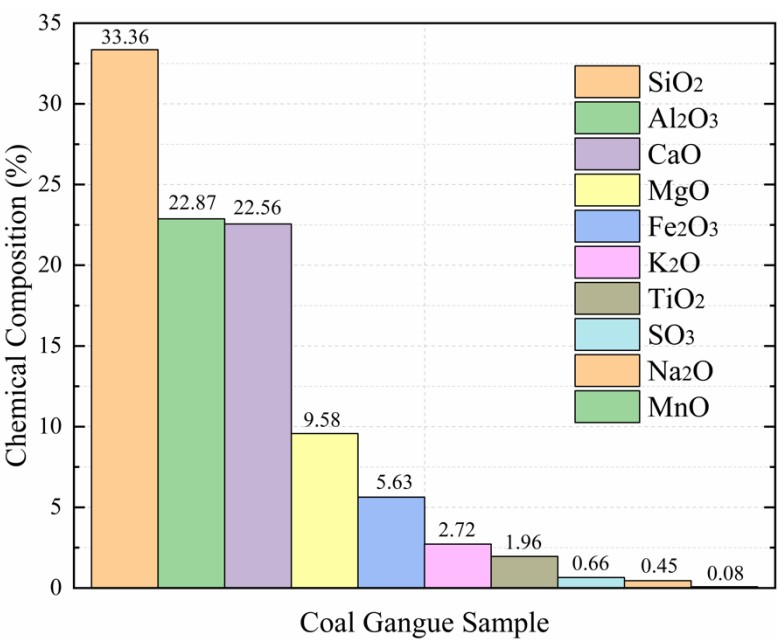

**Figure 3.** The chemical composition in the original coal gangue sample.

Figure 4d shows that the dissolved kalium curves fluctuated greatly and were quite complicated in comparison to the other major cations examined. The dissolved quantities of kalium showed significant variations, which maintained an overall N-shaped developing trend that was similar to the trends of other major cations examined. The dissolved kalium levels in the other solutions over time as the fourth major cations in water solutions were almost ordered as GK2 > GK1 > GU, as seen in Figure 3. The curve of GK1 fluctuated more greatly than that of GK2 in the initial stage (0–4 d), and it increased dramatically up to the peak value (18.2 mg/L), although the initial value was three times lower than that of GK2, and during this stage, it was only 5 mg/L less than that of GK2, and then decreased quickly to the value of 9.4 mg/L after 7 d, which was still lower than that of GK2. This indicated that the kalium concentrations of the solution GK1 experienced more violent reaction with the minerals from the coal gangue than that of the solution GK2 in the initial stage. However, the kalium curve trend of solution GU developed very similar to GK2 and maintained a lower level than that for other solutions, showing a relatively weak reaction process in kalium cation between the two solution and the coal gangue minerals. We also confirmed that the change curves in kalium concentrations for all immerse solutions almost had the same time schedule for the peak value and the minimum value of the kalium concentrations, indicating that the key influencing stage on kalium contents between the minerals in coal gangue and the water solutions may not matter much in relation to the water types.

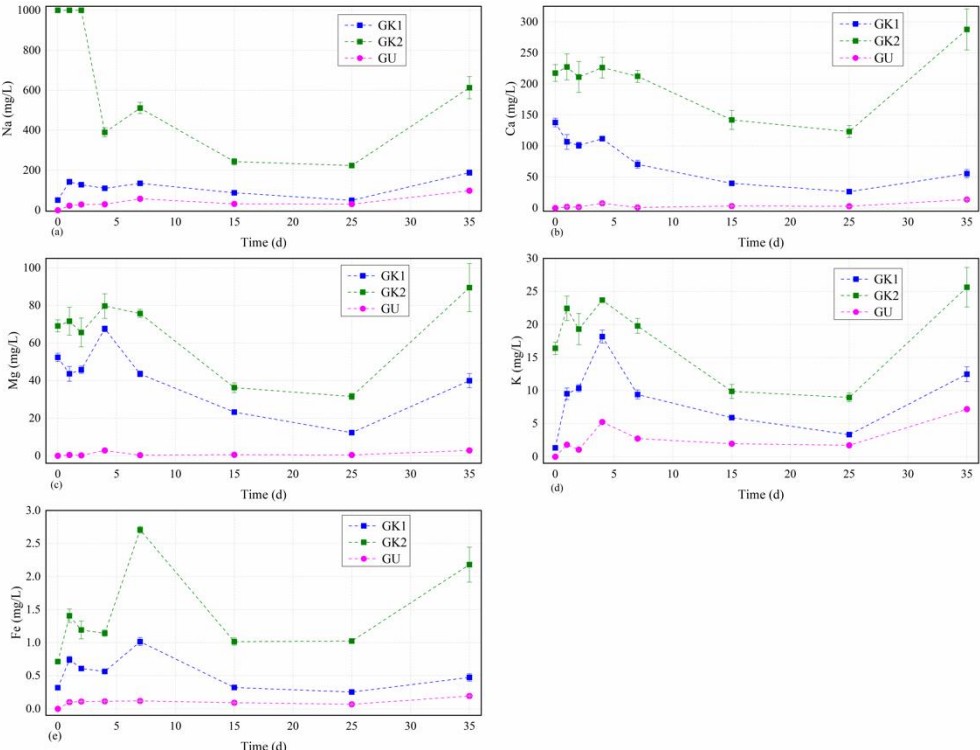

**Figure 4.** Concentrations' variation of several main cations in different immerse solutions over time. ((**a**) Trend plot of Na ions in immersion solution over time; (**b**).Trend plot of Ca ions in immersion solution over time; (**c**) Trend plot of Mg ions in immersion solution over time; (**d**) Trend plot of K ions in immersion solution over time; (**e**) Trend plot of Fe ions in immersion solution over time).

The dissolved Ca levels in the different solutions over time as the second major cations in water solutions were always ordered as GK2 > GK1 > GU, as seen in Figure 4b. The change trends of the Ca concentrations in GK2 and GK1 solutions were very similar, and the Ca concentrations fluctuated more and reached to the relatively lower level (227.3 mg/L and 106.6 mg/L, respectively) in the initial stage between 0 and 2 d, and then increased moderately to the little dissolved peak values (226.4 mg/L and 111.9 mg/L, respectively) after 4 d, three days earlier than the peak values of the Na concentrations curve; then, they decreased steadily to the almost lowest values (142.1 mg/L and 39.8 mg/L, respectively) after 15 d, similar to those of Na, had a slight downward trend to the lowest value (123.4 mg/L and 26.5 mg/L, respectively) between 15 and 25 d, and then finally increased to a relatively higher dissolved values (287.8 mg/L and 55.5 mg/L, respectively) between 25 and 35 d, whose values were obviously higher than their initial ones in solution GK2 but much less than the initial one in solution GK1. However, the Ca concentrations for solutions GU showed a similar developing trend with those of Na. In the initial stage (0–4 d), the Ca concentrations in solutions GU first gently rose up to the first dissolved peak value (7.8 mg/L) at the same peak time of the site karst water solutions, decreased slowly to the almost lowest values (1.2 mg/L) after 7 d, remained relatively stable after 7–25 d, and finally increased to the highest dissolved values (13.7 mg/L) after 35 d. From these results, we confirmed that the change curves in $Ca^{2+}$ for all solutions almost had the same time schedule for the peak value and the minimum value, indicating that the key influencing stage on Ca contents change may do not matter much with the water types.

The dissolved Mg change curves in all solutions over time as the third major cations in water solutions were extremely similar to those of Ca and corresponding with each other, having an order of GK2 > GK1 > GU, as seen in Figure 4c. However, the variation tendencies of Mg were much steeper than those of Ca, although the first dissolved peak values (79.7 mg/L, 67.6 mg/L, and 2.8 mg/L, respectively) of Mg in all immerse solutions were reached after 4 d and the lowest values (31.5 mg/L, 12.3 mg/L, and 0.4 mg/L,

respectively) were almost reached after 25 d, which were same as those of Ca. From these results, we also confirmed that the key influencing stage on Mg contents between the minerals in coal gangue and the water solutions do not matter much in relation to the water types.

The dissolved Fe levels in the other solutions over time were always ordered as GK2 > GK1 > GU, as seen in Figure 4e. The change curve of the Fe concentrations in site karst water solutions including GK1 and GK2 were very similar and maintained an overall M-shaped developing trend before 15 d, and the Fe concentrations in the two solutions increased up to the first little peak value (1.41 mg/L and 0.74 mg/L, respectively) in the initial stage between 0 and 1 d, decreased to the bottom value (1.15 mg/L and 0.56 mg/L, respectively) after 4 d, increased to the peak values (2.71 mg/L and 1.02 mg/L, respectively) after 7 d, decreased to the almost lowest values (1.02 mg/L and 0.32 mg/L, respectively) after 15 d, remained relatively stable in the middle stage between 15 and 25 d, and then finally increased moderately to a relatively higher dissolved values (2.18 mg/L and 0.48 mg/L, respectively) in the final stage between 25 and 35 d. Moreover, the change curve of the Fe concentrations in GK2 fluctuated much more greatly than the other two solutions, and the difference between solutions GK1 and GU was getting smaller on the whole, with the biggest difference of 0.90 mg/L after 7 d and the smallest one of 0.18 mg/L. However, the Fe concentrations in solution GU first increased up to 0.1 mg/L after 1 d, kept relatively stable between 1 and 7 d, and then decreased slowly to the lowest value of 0.07 mg/L after 25 d with a slightly increase until the end of the experiment. From these results, we confirmed that the change in Fe concentrations for GK1 and GK2 had more long fluctuations until 15 d than that for GU.

Through ion chromatograph (ICS), we also obtained the concentrations of main anions including those of sulphate, chloride, nitrate, and fluoride in different solutions over time depicted in Figure 4. Figure 4 presents that the concentration curves of all tested several anions in different solutions experienced a relatively larger fluctuation in the earlier stage (0–7 d) before keeping a relatively stable condition.

The dissolved sulphate levels in different solutions over time as the first major tested cations in water solutions were always ordered as GK2 > GK1 > GU, as seen in Figure 5a. The sulphate concentrations in different solutions over time showed an increasing trend as a whole, and the higher the initial concentration of sulphate in different solutions, the greater the fluctuation and the greater the increase. Each curve could be divided into two stages, including the initial fluctuation stage (0–7 d) and the stable stage (7–35 d). Solution GK2 as the owner of the highest initial sulphate concentration (1224.6 mg/L, much higher than the others) increased moderately up to the first little dissolved peak value (1368.4 mg/L) after 4 d, decreased quickly to a relatively lower level (1260.2 mg/L) after 7 d, and then slowly increased to the highest value (1375.2 mg/L) after 25 d with a final slightly downward trend to a relatively lower value (1320.4 mg/L) after 35 d with an increase by 0.1 times from beginning to end. GK1 solution as the owner of the third highest initial sulphate concentration (114.4 mg/L) basically remained unchanged (111–120 mg/L) before 2 d, slowly increased to the highest value (146.1 mg/L) after 7 d, and finally remained relatively stable. However, the dissolved sulphate concentrations in lab-configured water solutions (solutions GU) showed nonlinear growth starting from 0 over time, which were very different from that of site karst water, due to the continual dissolution of soluble minerals in coal gangue, with a final value of 49.9 mg/L after 35 d, respectively. The dissolved sulphate concentrations variations of the lab-configured water solutions (GU) over time were calculated by solving the fitting formulas, namely $C_{sul} = -0.034T^2 + 2.5391T + 14.916$, where $C_{sul}$ is the dissolved sulphate concentration of the lab-configured water solutions, and T is the immersion time. The correlation coefficient was 0.872. From these results, we confirmed that the key reaction stage on sulphate contents between the minerals in coal gangue and the site karst water solutions was still the initial stage (0–7 d). Although the earlier reaction was relatively stronger, it stabilized soon due to a dynamic balance of the interaction between the minerals in coal gangue and the original ions in the site

karst water. However, the dissolved sulphate concentration of the lab-configured water solutions was at a low level on the whole, indicating a different effect of site karst water and lab-configured water on the sulphate contents from the coal gangue, and it presented a very good nonlinear growth due to the oxidation of pyrite from the coal gangue.

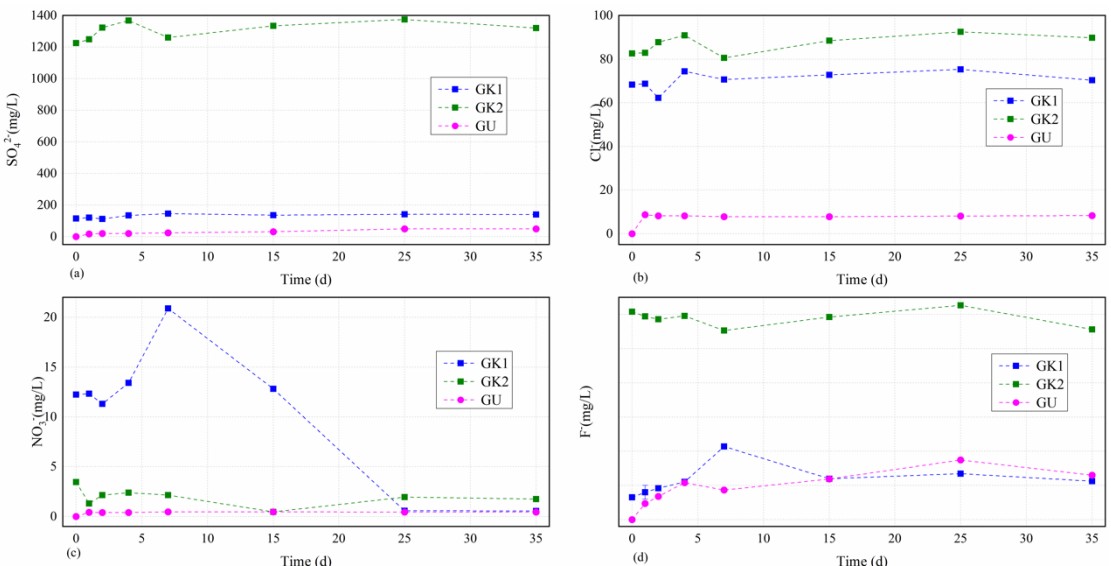

**Figure 5.** Concentrations' variation of several anions in different immerse solutions over time. ((**a**) Trend plot of Sulfate ions in immersion solution over time; (**b**) Trend plot of Chloride ions in immersion solution over time; (**c**) Trend plot of Nitrate ions in immersion solution over time; (**d**) Trend plot of Fluorine ions in immersion solution over time).

The dissolved chloride levels in site karst water solutions over time as the third major tested cations in water solutions basically maintained unchanged and mainly focused on the range of 70–90 mg/L, as seen in Figure 5b. The chloride concentrations in site karst water solutions over time showed a similar variation tendency and the biggest difference is the initial stage (1–2 d). The chloride concentrations in those solutions including GK2 and GK1 began to have a slight increase from 82.6 mg/L to 82.9 mg/L, and from 68.3 mg/L to 68.7 mg/L after 1 d, respectively. After that, a big difference that solution GK2 increased moderately to 87.8 mg/L, and solution GK1 increased dramatically to 62.3 mg/L occurred. Then, all of them experienced an increase to arrive at the first little dissolved peak value (90.9 mg/L and 74.4 mg/L, respectively) after 4 d, decreased moderately to a relatively lower level (80.5 mg/L and 70.6 mg/L, respectively) after 7 d, and then increased moderately to the highest value (92.5 mg/L and 75.3 mg/L, respectively), with a final decrease between 25 and 35 d. However, the dissolved chloride concentrations in lab-configured water solutions after 1 d experienced a dramatical respectively increase from 0 to 8.6 mg/L for solution GU. After that, the dissolved chloride concentrations in solution GU kept a stable range from 7.8 mg/L to 8.3 mg/L to the end of the experiment. From these results, we confirmed that the key reaction stage on chloride contents between the minerals in coal gangue and the site karst water solutions was still the initial stage (0–7 d). However, the dissolved chloride concentration of the lab-configured water solution was at a low level on the whole, similarly indicating a different effect of site karst water and lab-configured water on the chloride contents from the coal gangue.

The dissolved nitrate levels in site karst water solutions over time as the second major tested cations in water solutions experienced a very large fluctuation before 25 d, and finally maintained relatively unchanged after 25 d and were mainly focused on the range of 0.4–1.8 mg/L, as seen in Figure 5c. Moreover, the higher the initial concentration of nitrate in site karst water solutions, the greater the fluctuation of nitrate levels at the initial stage. GK1 solution, having the highest initial concentrations, experienced the largest fluctuation

throughout the experiment (from 20.9 mg/L to 0.6 mg/L), indicating a stronger reaction between the minerals from the coal gangue and the dissolved nitrate concentrations in GK1 solution than those of the others. GK1 solution first maintained relatively stable and then decreased from 12.3 mg/L to the first relatively lower level of 11.3 mg/L; after that, GK1 solution increased dramatically to the highest level of 20.9 mg/L before 7 d and then decreased quickly to the lowest level (0.6 mg/L) between 25 and 35 d. GK2 solution also experienced two depressions (1.3 mg/L after 1 d and 0.5 mg/L after 15 d) like GK1 solution, and it has two little peak value after 4 d and after 25 d with a relatively unchanged level of 1.8–1.9 mg/L between 25 and 35 d. However, the dissolved nitrate concentrations in lab-configured water solution, GU showed nonlinear growth starting from 0 over time like the sulphate level, which was very different from that of site karst water, due to the continual dissolution of soluble minerals in coal gangue, with a final value of 0.45 mg/L after 35 d, respectively. From these results, we confirmed that the key reaction stage on nitrate contents between the minerals in coal gangue and the site karst water solutions lasted longer than other anions focusing on the stage (0–25 d). However, the dissolved nitrate concentration of the lab-configured water solution was at a low level on the whole, indicating a different effect of site karst water and lab-configured water on the nitrate contents from the coal gangue. Moreover, the key reaction stage on nitrate contents between the minerals in coal gangue and the lab-configured water solution was in a single day.

The dissolved fluoride levels in site karst water solutions over time fluctuated relatively little overall and experienced a large fluctuation in solution GK1 in the initial stage between 0 and 7 d, and then remained relatively unchanged, as seen in Figure 5d. GK2 solution experienced the smallest fluctuation throughout the experiment (between 2.8 mg/L and 3.0 mg/L); GK1 solution increased moderately up to the highest value from 0.3 mg/L to 1.1 mg/L in the initial stage between 0 and 7 d, and then kept in the range of 0.6–0.7 mg/L, which is overall at a relatively lower level among all the anions. However, the dissolved fluoride concentrations in lab-configured water solution GU basically showed nonlinear growth starting from 0 to the highest value (0.9 mg/L) after 25 d over time, which were very different from that of site karst water, due to the continual dissolution of soluble minerals in coal gangue, with a final value of 0.6 mg/L after 35 d. From these results, we confirmed that the key reaction stage on fluoride contents between the minerals in coal gangue and the site karst water solutions still focused on the initial stage (0–7 d), but it was not strong compared with the other anions. Moreover, the dissolved fluoride concentration of the lab-configured water solution was at a relatively higher level on the whole, which was even higher than that of GK1 solution, indicating that the fluoride in water may come from the minerals in coal gangue.

Cations of different immersion solutions have a similar trend of change, increasing and decreasing before soaking for 25 days, and gradually increasing after 25 days. $Na^+$ concentration fluctuation in the karst water immersion solutions system is more severe than that in the ultrapure water immersion solutions system, which is related to higher initial sodium ions concentration (common ion effect), ions exchange, and precipitation. The changes of calcium and magnesium ions in karst water are similar, with obvious fluctuation in soaking early stage and slow fluctuation in soaking middle and late stage, which is less affected by the initial ions concentration in karst water.

However, the anions showed a similar trend, with significant fluctuation only in the early stage of soaking and small fluctuation in the late stage. Compared with ultrapure water immersion solution, sulfate ions fluctuation is more obvious in karst water immersion, which is affected by the initial ions in karst water. $NO_3^-$ fluctuates longer than $SO_4^{2-}$, and nitrate ion concentration in GK1 fluctuates sharply, which is due to the higher initial $NO_3^-$ in GK1 that enhances the ions interaction. The $Cl^-$ concentration of GK1 and GK2 immersion solutions showed a similar variation tendency, and there was little change throughout the experiment. Solutions with lower initial $F^-$ ions concentration have more violent fluctuations, while those with higher $F^-$ ions concentration have less fluctuations.

### 3.3. Correlation of Main Hydrochemical Indexes and Main Controlling Factors

By adopting Origin 2021 software, the 100% stacked column graphs among the main hydrochemical ions variation (Na, K, Ca, Mg, Fe, sulfate, chlorides, nitrate, and fluoride), the difference from beginning to end in all water solutions can be obtained, which is shown in Figure 5.

As can be seen in Figure 6, in terms of the concentration proportion of each component in the three immersion solutions, sodium, sulfate, chlorides, calcium, and potassium ions always dominate in GU immersion solution. In terms of concentration contribution rate alone, Na and $SO_4^{2-}$ ions account for 67–86% of the total, with a small fluctuation range; secondly, Cl, Ca, K, and Mg ions account for only 14–24%, and the contribution rate of other components is low. In GK1 immersion solution, Na, $SO_4^{2-}$, Ca, Cl, and Mg ions always dominate. In terms of concentration contribution rate alone, Na and $SO_4$ account for 38–65% of the total, with a large fluctuation range, especially in the initial stage, accounting for the smallest proportion; secondly, Ca, Cl, and Mg ions account for 33–59%. At the initial stage, Ca ions account for the largest proportion, surpassing all other ions, and the contribution rate of other components is low. $SO_4^{2-}$, Na, Ca, Cl, and Mg ions always dominate in GK2 immersion solution. In terms of concentration contribution rate alone, Na and $SO_4^{2-}$ account for 81–86% of the total and are the most stable among the three solutions; secondly, Ca, Cl, and Mg ions account for 13–20%, and the contribution rate of other components is relatively low. GK2 is the most stable component in the three solutions. Among the three kinds of aqueous solutions, the hydrochemical components of high salinity immersion solutions, especially those dominated by Na and $SO_4^{2-}$ ions, have basically reached dynamic equilibrium, and are less affected by coal gangue minerals, which plays a key role in the balance of the ionic components of the solution. Compared with ultrapure water solution, for the aqueous solution with a certain concentration in the initial state, such as GK1 immersion solution, under the action of coal gangue, its ionic components have undergone a relatively strong recombination process.

From the TDS, pH, and ORP of the three immersion solutions, the total dissolved solids in GU immersion solution increased to 120.47 mg/L, an increase of nearly 4154.02%, and the pH in the solution increased by nearly 45%; the oxidation environment changed into the reduction environment, with the greatest degree of change. The total dissolved solids in GK1 immersion solution decreased by 96 mg/L, nearly 19.98%, and the pH in the solution increased by 11.52%; the redox environment changes minorly. The total dissolved solids in GK2 immersion solution increased to 61.97 mg/L, an increase of nearly 4.95%, and the pH change in the solution was the smallest, an increase of only 8.06%; the redox environment changed minorly. By comparing the three indicators in the immersion solution, it is found that GK2 immersion solution is the most stable of the three immersion solutions. It can be seen that the high salinity mine water solution plays a good buffer role, making the basic water quality indicators in the immersion solution, such as TDS, pH, and ORP, change the least; the ultrapure water GU with the lowest salinity is the largest in terms of total dissolved solids, pH, or ORP. Therefore, for the immersion solution with low ion concentration, the change of water quality index of the immersion solution after the interaction between coal gangue and water mainly depends on the dissolution and release of minerals in coal gangue.

In terms of time scale, the minerals in coal gangue in GU immersion solution are rapidly dissolved and released in 0–1 days and form the initial equilibrium state of each ion component, the relative scale of each ion component is in a relatively stable period within 1–2 days, and the relative scale of each ion changes slightly; in 4–36 days, the relative scale of each ion component is in the fluctuation equilibrium period, and the relative scale of each ion changes greatly, but it returns to stability on the last day. In the fluctuation equilibrium period, the main changing ions in the aqueous solution are Na and $SO_4^{2-}$ ions, followed by Ca ions, and the concentration of Cl ions remains stable as a whole; the main period of ion exchange between Ca, K, and Mg ions and Na ions is 2–7 days; Ca/K/Mg ions are exchanged into Na ions in 1–2 days, Na ions are exchanged into Ca/K/Mg ions in 2–4 days,

and Ca/K/Mg ions are exchanged into Na ions in 4–7 days. For GU immersion solution, the ion exchange capacity is the largest in the first 7 days, and the proportion of each ion in the solution enters the first relative scale equilibrium period on the 7th day. Combined with the change of ion concentration in 7–15 days, it is found that the cation exchange capacity between Na ion concentration and Ca/K/Mg ions is very small; the reason may be that, on the 7th day, the concentration of Na ions in the immersion solution basically reached saturation, and the concentration difference between the Na ions in the solution and the Na ions in the colloidal particles of coal gangue formed like the internal and external environment of the cell. It began to enter the pores of coal gangue and adsorbed on the surface of colloidal particles, resulting in a significant reduction in the concentration of Na ions in the immersion solution. When the colloidal particles adsorbed in coal gangue reach saturation, its adsorption capacity will slowly decrease and be released into the immersion solution, with the continuous dissolution of coal gangue minerals, a new concentration equilibrium is formed; that is, in 7–35 days, the GU immersion solution enters the second relative scale equilibrium period, and this process will continue to repeat in the future soaking cycle.

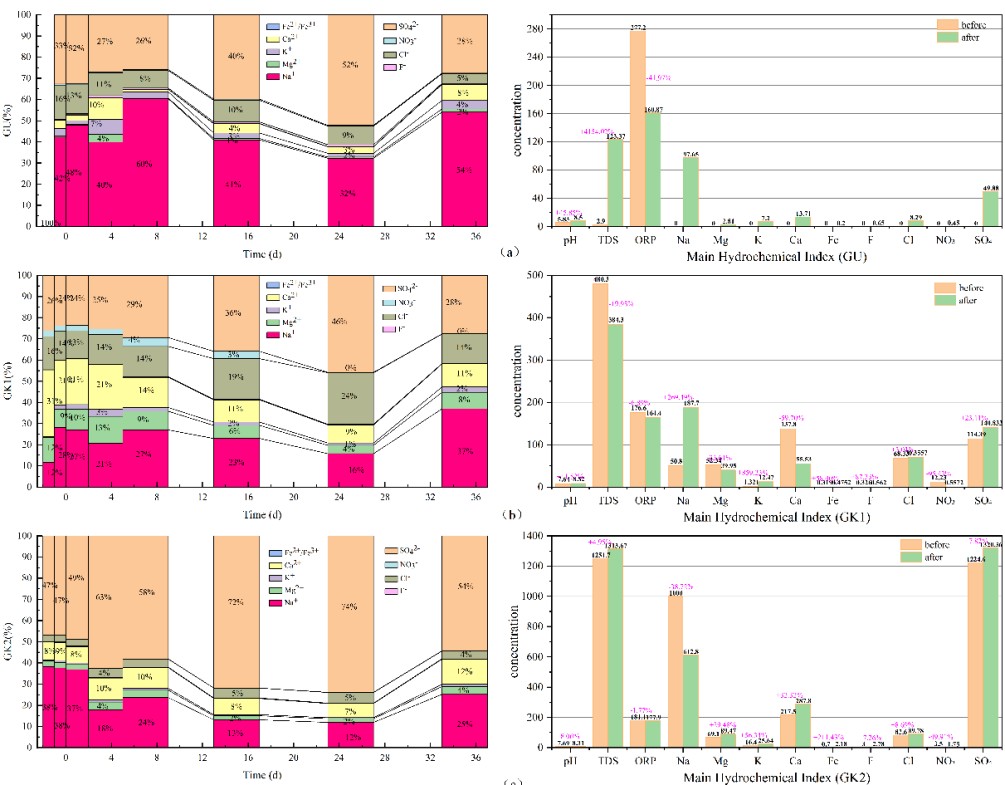

**Figure 6.** The main hydrochemical ions variation relationship diagram throughout the experiment and the difference from beginning to end. ((**a**) Graph of the proportion change of water chemical ions in GU immersion solution and the difference in the data; (**b**) Graph of the proportion change of water chemical ions in GK1 immersion solution and the difference in the data; (**c**) Graph of the proportion change of water chemical ions in GK2 immersion solution and the difference in the data).

In 0–1 days, the concentration of each ion component in GK1 immersion solution rapidly recombines from the initial equilibrium state. When the concentration of Na is low, Ca and Mg ions exchange with Na ions, and the relative scale of each ion component is in a relatively stable period within 1–2 days, with little change in the relative scale of each ion; in 4–36 days, the relative scale of each ion component is in the fluctuation equilibrium period, and the relative scale of each ion changes greatly, but it returns to stability on the last day. In the fluctuation equilibrium period, the main changing ions in the immersion solution are Na, Ca, and $SO_4^{2-}$ ions, followed by Mg and K ions, and the concentration

of Cl ions remains stable as a whole; consistent with the interaction process of GU water coal gangue samples, the main period of ion exchange between Ca, K, and Mg ions and Na ions is 2–7 days. In 1–2 days, Ca and Mg ions are exchanged into Na and K ions; in 2–4 days, Na ions are exchanged into Ca/K/Mg ions; and in 4–7 days, Ca/K/Mg ions are exchanged into Na ions. For GK1 aqueous solution, the ion exchange capacity is the largest in the first 7 days, and the proportion of each ion in the solution enters the first relative scale equilibrium period on the 7th day. Combined with the change of ion concentration on days 7–15, it is found that the cation exchange capacity between Na ion concentration and Ca/K/Mg ions is very small, and in 7–35 days, the GK1 immersion solution enters the second relative scale equilibrium period.

In 0–1 days, the concentration of each ion component in GK2 aqueous solution rapidly recombines from the initial equilibrium state. Under the condition of high Na concentration, the possibility of cation exchange is very small, and other ions are in an increasing trend. Although Na ions also exchange with Ca, Mg, and K ions, they does not play a leading role, and the relative scale of each ion component is in a relatively stable period within 1–2 days, with small changes in the relative scale of each ion. From 4 to 36 days, the relative scale of each ion component is in the fluctuation equilibrium period, and the relative scale of ions changes greatly, but it returns to stability on the last day. In the fluctuation equilibrium period, the main changing ions in the aqueous solution are Na, $SO_4^{2-}$, and Ca ions, followed by Mg and K ions, and the concentration of Cl ions remains stable as a whole. Due to the high concentration of Na ions and $SO_4^{2-}$, the ion exchange between Ca, K, and Mg ions and Na ions is very small. In the first 7 days, the GK2 immersion solution is significantly different from the first two types of water samples, but the relative scale of each ion in the solution enters the first concentration equilibrium period on the 7th day. Combined with the change of ion concentration from 7 to 15 days, it is found that the cation exchange between Na ion concentration and Ca, K, and Mg ions is very small; in 7–35 days, GK2 immersion solution enters the second relative scale equilibrium period, which is generally consistent with the action process of the first two types of water samples. Compared with other cations, the Na ion has a weak ability to adsorb on the surface of colloidal particles. With the change of Na ion concentration, the Na ion on the surface of colloidal particles is more likely to be lost to the aqueous solution, so its concentration changes violently in the whole test process.

By analyzing the dominant anions $SO_4^{2-}$ and Cl ions in the three kinds of immersion solutions, it can be found that although Cl ions basically do not change greatly after reaching the saturation solubility, the relative scale of each ion component is still violently recombined and then enters the relative scale equilibrium period again. It can be seen from Gu aqueous solution that the rapid increase of $SO_4$ ion concentration may be mainly formed by the oxidation of sulfide in coal gangue particles, and the major change of its redox environment is its important evidence.

In addition, $SO_4^{2-}$ ions have a similar change law with the dominant cation Na. Comparing three kinds of aqueous solutions, it can be found that in the first 7 days, the concentration of Na ions entering the surface of colloidal particles inside the coal gangue is small, which still belongs to the accumulation process. In the 7–36 days of the fluctuation equilibrium period, when the Na ions adsorbed on the surface of colloidal particles of coal gangue reach a certain concentration, it will cause the loss of $SO_4^{2-}$ ions, and the negative correlation between $SO_4^{2-}$ ions and Na ions is more obvious, forming the alternating adsorption of cation and anion.

After the water–rock reaction, the mine water quality type changes, and the water quality type changes as shown below (Figure 7 and Table 2).

According to the above analysis, Ca and Mg ions and Na in GK1 immersion solution (within 2–7 d) are exchanged. and the exchange effect is strong in the early leaching stage; in addition, $SO_4^{2-}$ and $Cl^-$ do not change much after reaching saturation solubility, so the water quality type of GK1 after leaching changes from SO4-Cl-Na-Ca type to SO4-Cl-Na-Mg type (as shown in Figure 7 and Table 2), and the water quality type changes. Although

ion exchange occurs in the early stage of leaching in GK2, the exchange effect is not strongly correlated, and the $SO_4^{2-}$ ions in the anion always occupy a significant advantage, so the water quality type does not change after GK2 leaching.

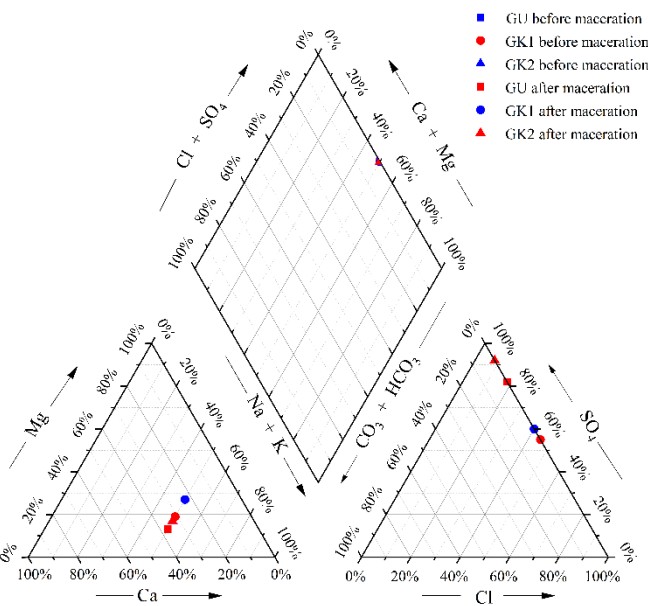

**Figure 7.** Piper three-line diagram of mine water before and after leaching.

**Table 2.** Type of mine water quality before and after leaching.

| Water Sample Number | Type of Water Quality before Leaching | Type of Water Quality after Leaching |
|---|---|---|
| GU | | SO4-Na-Ca |
| GK1 | SO4-Cl-Na-Ca | SO4-Cl-Na-Mg |
| GK2 | SO4-Na-Ca | SO4-Na-Ca |

## 4. Conclusions

(1) In the immersion solution with high mineralization, especially the aqueous solution dominated by Na and $SO_4^{2-}$ ions, its water chemical components have basically reached a dynamic equilibrium, and the influence of gangue minerals is small, which plays a key role in the balance of the ion components of the solution.

(2) According to the changes in the three water quality indicators in the three groups of aqueous solutions, it was found that the water quality indicators in the aqueous solution of GK2 were the most stable. Reflecting the high mineralization of the mine aqueous solution has a good buffer effect on the changes in water quality indicators such as pH, TDS, and ORP, the changes in water quality indicators are small. In aqueous solutions with low ion concentrations, the changes in water quality indicators are more drastic, and the changes in water quality indicators are mainly released by the dissolution of minerals in gangue.

(3) According to the curve of the change of ion concentration in each immersion solution, the three stages of the interion reaction were obtained, including the initial violent fluctuation stage (0–7 days), the intermediate stabilization stage (7–25 days), and the final slow reaction stage (25–35 days). In the early stage of leaching (0–7 d), sodium ions showed the opposite trend of other cations, the ion exchange reaction was stronger and then entered the relative scale equilibrium period, and the amount of ion exchange decreased.

**Author Contributions:** Writing-review and editing, D.-J.X.; Conceptualization, B.-B.J.; methodology, Y.-K.H.; formal analysis, Z.-G.C.; software, M.W. All authors have read and agreed to the published version of the manuscript.

**Funding:** The authors also gratefully acknowledge the projects of Open Fund of State Key Laboratory of Water Resource Protection and Utilization in Coal Mining (Grant No.WPUKFJJ201909), the Natural Science Foundation of Shandong Province, China (ZR2022MD101), and Science and Technology Research Directive Plans of China National Coal Association (Grant MTKJ2018-263).

**Data Availability Statement:** The data presented in this study are available on request from the corresponding author. The data are not publicly available due to confidentiality.

**Acknowledgments:** Our deepest gratitude goes to the editors and anonymous reviewers for their careful work and thoughtful suggestions that have helped to substantially improve this paper.

**Conflicts of Interest:** The authors declare no competing financial interest.

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
