# Peer review of "Effects of Site Karst Water on Fresh Coal Gangue at Baizhuang Coalfield, China—Leaching Characteristic"

_water, doi:10.3390/w14203267_

Round 1
Reviewer 1 Report
This article is devoted to a very important problem - the study of changes of groundwater quality in the zone of mining coal deposits. The results of the laboratory experiments are described in detail. The changes of separated anions and cations are presented at the figures 3-5. Unfortunately, a comprehensive analysis of chemical composition changes in the water samples has not been performed. For example, you can judge about general trend in the chemical composition using a triangle graph (see Appendix - Figure 1). A transition from the Ca2+ type across mixed type to the Na++K+ was traced for GK1 fresh water. The cationic composition of GK2 weakly brackish water remains Na++K+. A similar analysis or other types of complex data processing should be carried out to estimate of anions concentration. Information about the methods error for ions content determining is also absent in article. It is not explained why the bicarbonates, carbonates, gases concentration and others parameters were not determined. It is not clear why the temperature of 40 degrees is chosen as the main one. It is recommended to improve this article.
Author Response
Dear Reviewer or Editor:
Thank you for the comments concerning our manuscript, No. water-1868914. Those comments are all valuable and very helpful for revising and improving our paper, as well as the important guiding significance to our research. We have studied comments carefully and have made correction which we hope meet with approval. Furthermore, according to Water Editor’s suggested, we to the main corrections in the paper and the responds to the reviewer’s comments are as follows:
Responds to Reviewer #1:
- Response to comment: (Unfortunately, a comprehensive analysis of chemical composition changes in the water samples has not been performed. For example, you can judge about general trend in the chemical composition using a triangle graph (see Appendix - Figure 1). A transition from the Ca2+ type across mixed type to the Na++K+ was traced for GK1 fresh water.)
Response: Thank you very much for your comment, according to your comment, we have added Piper three-line diagram for judgment of the overall chemical composition trends and data analysis.
- Response to comment: (Information about the methods error for ions content determining is also absent in article.)
Response: Thank you very much for your comment, according to your comment, we have modified the information about the methods for ions content determining.
- Response to comment: (It is not explained why the bicarbonates, carbonates, gases concentration and others parameters were not determined.)
Response: Thank you very much for your comment, because the test progress was frequent, according to the actual experimental situation, only part of the ions was tested, and many ions content (such as HCO3- ions) was not determined.
- Response to comment: (It is not clear why the temperature of 40 degrees is chosen as the main one.)
Response: Thank you very much for your comment, in order to realistically simulate the on-site temperature conditions of the goaf, the temperature of 40 degrees is chosen as the main one during the experiment. We have stated this at "2.2 Static Immersion Test".
We tried our best to improve the manuscript and made some changes in the manuscript. These changes will not influence the content and framework of the paper. And here we did not list the changes throughout the paper.
We appreciate for Editor/Reviewers’ warm work earnestly, and hope that the correction will meet with approval.
Once again, thank you very much for your comments and suggestions.
Yours sincerely,
Xu, Prof.
Reviewer 2 Report
The authors in the manuscript entitled "Effects of Site Karst Water on Fresh Coal Gangue at Baizhuang Coalfield, China: Leaching Characteristic" have studied the effects of gangue on water quality change under different immersion solution conditions. It is been presented well. Basically, this study attempted to provide a theoretical basis for exploring the changes of gangue to the quality of karst water in Ordovician limestone and also looking forward to more studies on groundwater pollution mechanisms in closed coal mines.
Here are few suggestions to enhance the quality of the manuscript:
1. The introduction section has been well written although there are a few grammatical errors.
2. Picture 2 is unclear. Please provide a high-resolution image. In pictures 3 and 4, for GU, instead of brown color dots please use another intense color.
3. Picture 5 is also blurring. The levels on X-axis and Y-axis are hard to read. Please completely modify the pictures.
4. An graphical abstract for the study is important. It allows the readers to quickly gain an understanding of the content of the study. Please provide a graphical abstract.
Author Response
Dear Reviewer or Editor:
Thank you for the comments concerning our manuscript, No. water-1868914. Those comments are all valuable and very helpful for revising and improving our paper, as well as the important guiding significance to our research. We have studied comments carefully and have made correction which we hope meet with approval. Furthermore, according to Water Editor’s suggested, we to the main corrections in the paper and the responds to the reviewer’s comments are as follows:
Responds to Reviewer #2:
- Response to comment: (1. The introduction section has been well written although there are a few grammatical errors.)
Response: Thank you very much for your comment, according to your comment, we have modified the grammatical errors.
- Response to comment: (2. Picture 2 is unclear. Please provide a high-resolution image. In pictures 3 and 4, for GU, instead of brown color dots please use another intense color.)
Response: Thank you very much for your comment, according to your comment, we provide a high-resolution image of Figure 2. In addition,we used another intense color in figure 3 and 4 for express GU.
- Response to comment: (3. Picture 5 is also blurring. The levels on X-axis and Y-axis are hard to read. Please completely modify the pictures.)
Response: Thank you very much for your comment, according to your comment, we provide a more clear picture, Figure 5.
- Response to comment: (An graphical abstract for the study is important. It allows the readers to quickly gain an understanding of the content of the study. Please provide a graphical abstract.)
Response: Thank you very much for your comment, according to your comment, we provide a graphical abstract for this essay.
We tried our best to improve the manuscript and made some changes in the manuscript. These changes will not influence the content and framework of the paper. And here we did not list the changes throughout the paper.
We appreciate for Editor/Reviewers’ warm work earnestly, and hope that the correction will meet with approval.
Once again, thank you very much for your comments and suggestions.
Yours sincerely,
Xu, Prof.
Reviewer 3 Report
Please see the attached document

Author Response
Dear Reviewer or Editor:
Thank you for the comments concerning our manuscript, No. water-1868914. Those comments are all valuable and very helpful for revising and improving our paper, as well as the important guiding significance to our research. We have studied comments carefully and have made correction which we hope meet with approval. Furthermore, according to Water Editor’s suggested, we to the main corrections in the paper and the responds to the reviewer’s comments are as follows:
Responds to Reviewer #3:
- Response to comment: ((1) English in the manuscript needs to be substantially improved;)
Response: Thank you very much for your comment, according to your comment, we have modified English quality for the manuscript.
- Response to comment: ((2) The concept of specific gravity is not correctly used;)
Response: Thank you very much for your comment, according to your comment, we have modified "specific gravity" to be "relative scale".
- Response to comment: ((3) What is called wave equilibrium period?)
Response: Thank you very much for your comment, according to your comment, we have modified "wave equilibrium period" to be "fluctuation changes period".
- Response to comment: ((4) In order to study the influence of coal gangue on the quality of groundwater, the authors have to know what minerals in coal gangue, which control the chemical reactions with water, and further change water quality. Only knowing the oxides in coal gangue is not enough to explore the investigation that the manuscript set up. The mineralogy can be known by XRD and petrographic study by optical microscopy, even by electric microscope.)
Response: Thank you very much for your comment, according to your comment, we added the XRD test results of minerals to explain the mineral components contained in coal gangue.
- Response to comment: ((5) Without considering minerals, such an investigation cannot gain any insights into how and why water quality changed;)
Response: Thank you very much for your comment, according to your comment, we have added the content about the mineral components of coal gangue.
- Response to comment: ((5) Why did the authors choose to only use the part of the gangue sample with diameter size of 0.45~3.2 mm?)
Response: Thank you very much for your comment, according to some scholars' research results about the proportional relationship between the coal gangue and the restored mine water, it is easier to select coal gangue with a diameter size 0.45~3.2mm for experimentation.
- Response to comment: ((6) Why did the authors take the sampling time at 1 d, 2 d, 4 d, 7d, 15d, 25d, and 35d during their experiment?)
Response: Thank you very much for your comment, we selected the 1,2,4,7,15,25, and 35d times were set to explore water quality changes at longer time intervals.
- Response to comment: ((7) Why did the authors take the sampling time at 1 d, 2 d, 4 d, 7d, 15d, 25d, and 35d during their experiment?)
Response: Thank you very much for your comment, we selected the 1,2,4,7,15,25, and 35d times were set to explore water quality changes at longer time intervals.
- Response to comment: ((8) How did the authors control the temperature, pressure and oxygen fugacity in their experiment? How much uncertainty with them?)
Response: Thank you very much for your comment, in order to realistically simulate the on-site temperature conditions of the goaf, we used the 40°C constant temperature incubator keep the temperature constant.
- Response to comment: ((9) Is there any sulfides in coal gangue? The increased sulfate concentration implies there may be some sulfides in gangue.)
Response: Thank you very much for your comment, XRD mineral analysis indicates the presence of sulfate minerals in coal gangue. We infer the increased sulfate concentration associated by sulfide dissolution.
- Response to comment: ((10) The concentration of nitrate changed in a very large range indicates that microbial related reaction cannot be ignored. That means the experiment was influenced by microbial community.)
Response: Thank you very much for your comment, because the experimental conditions are limited, all the influencing factors cannot be considered, and the experimental process did not consider the influence of the microbial community.
- Response to comment: ((11) Why didn't the authors show the changes of pH, TDS and ORP with time?)
Response: Thank you very much for your comment. Limited by the length of the article, combine with the topic of the article, the main analysis of the changes in the composition of water chemistry, the article only carried out a review of pH, TDS and ORP changes, and did not carry out specific change analysis.
We tried our best to improve the manuscript and made some changes in the manuscript. These changes will not influence the content and framework of the paper. And here we did not list the changes throughout the paper.
We appreciate for Editor/Reviewers’ warm work earnestly, and hope that the correction will meet with approval.
Once again, thank you very much for your comments and suggestions.
Yours sincerely,
Xu, Prof.
Round 2
Reviewer 1 Report
The authors of this article took into account the reviewer comment and made the appropriate change. The article may be published.
Reviewer 2 Report
The modifications made to the manuscript are satisfactory. Thank you.
Author Response
Thank you for the comments concerning our manuscript.
Reviewer 3 Report
Please see the attachment file

Author Response
Dear Reviewer or Editor:
Thank you for the comments concerning our manuscript, No. water-1868914. Those comments are all valuable and very helpful for revising and improving our paper, as well as the important guiding significance to our research. We have studied comments carefully and have made correction which we hope meet with approval. Furthermore, according to Water Editor’s suggested, we to the main corrections in the paper and the responds to the reviewer’s comments are as follows:
Responds to Reviewer #3:
1.Response to comment: (As karst water, it is surprised that they didn’t measure alkalinity or carbonate and bicarbonate for the initial water and during leaching experiment;)
Response: Thank you very much for your comment, because the test progress was frequent, according to the actual experimental situation, only part of the ions was tested, and many ions content (such as HCO3- ions) was not determined.
2.Response to comment: (For their leaching experiment, they only provided the chemical compositions of the coal gangue, but didn’t give the quantitative mineralogy except qualitative mineral information (even that XRD diagram, i.e. Fig.3, they didn’t mention it was done for their sample or from others). As an initial point of the experiment, if they don’t know exact quantitative mineralogy and chemical composition of each minerals in coal gangue, the leach experiment would have an half failed.)
Response: Thank you very much for your comment, according to your recommendations, we added quantitative results of minerals based on XRD analysis.
3.Response to comment: (pH is always an important parameter for any water-related experiment. The authors could not provide the diagram of pH changing with time during their leaching experiment, which I required in the first review.)
Response: Thank you very much for your comment. The article details the pH trends on 3.3 chapter, please check them out.
4.Response to comment: (Because of lacking quantitative mineralogy, the authors didn’t explain the leaching experiment reasonably well; because of lacking quantitative mineralogy, they cannot use hydrogeochemical modeling to explain their results.)
Response: Thank you very much for your comment, according to your recommendations, we added quantitative results of minerals based on XRD analysis.
5.Response to comment: (Incomplete experimental design result in unexplained results.)
Response: Thank you very much for your comment. In the future, when designing experiments, we will pay more attention to structural integrity, thank you!
6.Response to comment: (The author had not told yet in their coal gangue whether there is any sulfide like pyrite, and there is any potential contaminants like Pb, Zn, Cd, Cr, etc.)
Response: Thank you very much for your comment. According your opinion, we have added the corresponding references that trace amounts of pyrite are contained in gangue, but the XRD test failed to detect it because the pyrite content was less than 3%.
The chemical characteristics of groundwater water itself is a very complex process, it is impossible to use an experiment to take into account all the problems.
As far as possible, the revision has been made in accordance with the requirements of the reviewer, these changes will not influence the content and framework of the paper. And here we did not list the changes but marked in the paper.
We appreciate for Editor/Reviewers’ warm work earnestly, and hope that the correction will meet with approval.
Once again, thank you very much for your comments and suggestions.
Yours sincerely,
Xu, Prof.